# OpenReview forum: "AutoDefense: Multi-Agent LLM Defense against Jailbreak Attacks"
_NeurIPS.cc/2024/Workshop/SafeGenAi — SafeGenAi Poster_

### Official Review · Reviewer_3XSm · 2024-10-08
**Multi-agent framework for jailbreak defense, a good follow-up work of autogen**

**Rating:** 6
**Confidence:** 3

**Review:**

This paper introduces a multi-agent framework called AutoDefense, designed to defend large language models (LLMs) from jailbreak attacks. Jailbreak attacks manipulate LLMs into producing harmful or inappropriate content, bypassing built-in safety mechanisms. The AutoDefense system employs multiple LLM agents, each responsible for a specific defense task, to collaboratively filter and block harmful responses. The framework allows for flexibility and supports integration with other defense components.

---
## Strength
- **Novel Multi-Agent Defense Approach:** The paper introduces an innovative multi-agent framework that assigns distinct roles to LLM agents, improving the defense against jailbreak attacks by enhancing collaborative decision-making.
- **Flexibility and Expandability:** AutoDefense is designed to be modular, allowing for integration with other defense mechanisms, such as Llama Guard, which significantly improves its performance.
- **Strong Experimental Results:** The paper demonstrates a substantial reduction in the attack success rate (ASR) across various LLMs, with AutoDefense outperforming existing methods, notably reducing GPT-3.5's ASR from 55.74% to 7.95%.

---
## Weakness
- **Potential Computation Overhead:** The multi-agent setup, particularly with configurations involving multiple agents, could introduce notable computational costs and delays in real-time applications. Therefore, I think this method lacks real-world application due to its computational burden.

---
## Suggested Improvement:
- Table 2 exceeds the linewidth, the authors are suggested to change the format.
- Add a takeaway in the introduction section for how to use this method to balance efficiency and effectiveness.
- Although limitations remain, this paper is qualified for acceptance.

---

### Official Review · Reviewer_tCjM · 2024-10-09
**multi-agent defense framework for mitigating jailbreak attacks**

**Rating:** 6
**Confidence:** 3

**Review:**

The authors presented AutoDefense, a multi-agent defense framework for mitigating jailbreak attacks on large language models (LLMs).

cons: Tables 2 and 3 need to be aligned within the page margins. Currently, they extend outside the page boundaries.

---

### Official Review · Reviewer_Gf1f · 2024-10-10

**Rating:** 5
**Confidence:** 5

**Review:**

1. **Quality**: The quality of the research is moderate.


   **a)** The methodological framework is clear but lacks depth in its design. The authors have merely assigned different system prompts to LLMs and sequentially invoked the intention analyzer, prompt analyzer, and judge.


   **b)** The experimentation is insufficient. In Table 6, the authors tested the effectiveness of three types of prompt-level jailbreak attacks (and their combinations) on four different LLMs (GPT-3.5, Vicuna-13b, LLaMA-2-70b, and mixtral-8x7b). Thus, in Section 5.1, each victim model was tested with only one type of attack method. On one hand, all selected attack methods are prompt-level jailbreak attacks, which lacks results from token-level attacks [1-3], failing to demonstrate the defense method's effectiveness and generalization capability. On the other hand, comparisons with post-finetuning defense methods like Adversarial Training are missing.


   **c)** According to Table 4, ASR actually increases with the number of agents, such as when using GPT-3.5 and mixtral-8x7b as the defense agent backbone. Other metrics such as FPR and Accuracy are not necessarily best with three agents, failing to showcase the advantages of the multi-agent approach.


   **d)** In the multi-agent system, it appears that the backbone LLM of each agent is the same. The authors might consider using different LLMs for different components. For instance, employing GPT-4 as the analyzer for better analysis ability and Llama Guard 3 as the judge for more accurate classification.


[1] Zou, Andy, et al. "Universal and transferable adversarial attacks on aligned language models." arXiv preprint arXiv:2307.15043 (2023).

[2] Liao, Zeyi, and Huan Sun. "Amplegcg: Learning a universal and transferable generative model of adversarial suffixes for jailbreaking both open and closed llms." arXiv preprint arXiv:2404.07921 (2024).

[3] Paulus, Anselm, et al. "Advprompter: Fast adaptive adversarial prompting for llms." arXiv preprint arXiv:2404.16873 (2024).



2. **Clarity**: The paper is generally well-organized, and sections are logically structured. The figures and tables are informative and enhance the understanding of the text. However, some sections, particularly in the experimental setup (Section 4) and experimental results (Section 5), could benefit from more concise language and better explanations of the defense and attack baselines used in the experiments.


3. **Originality**: The concept of using a multi-agent system for LLM defense is novel. However, the method described in this paper does not truly represent a multi-agent system. It lacks a communication mechanism design among agents for jailbreak defense and does not demonstrate the advantages of agents being able to use tools or interact with the environment.


4. **Significance**: The significance of the work is potentially high given the increasing importance of securing LLMs against adversarial attacks. However, the practical applicability of the system may be limited by the defense performance and limited generalization ability, as well as computational overhead—issues that are not adequately addressed in the paper.


5. **Typos**:

a) In Table 2, "GPT35Turbo" should be corrected to "GPT-3.5-Turbo."

b) In Table 7, the ASR for CoT is the best for Vicuna-13B, so the result in the second row should be bolded, not the fourth row.

c) In line 135, "figure 4" should be corrected to "Figure 4."